# A Multi-Site Refinement Study of *Taking Back Control Together*, an Intervention to Support Parents Confronted with Childhood Cancer

**DOI:** 10.3390/curroncol32050253

**Published:** 2025-04-26

**Authors:** Nikita Guarascio, Ariane Levesque, David Ogez, Valérie Marcil, Daniel Curnier, Véronique Bélanger, Émélie Rondeau, Katherine Péloquin, Caroline Laverdière, Raoul Santiago, Josée Brossard, Stéphanie Vairy, Serge Sultan, the TBCT-Québec Team

**Affiliations:** 1Department of Psychology, Université de Montréal, Montreal, QC H2V 2S9, Canada; 2Azrieli Research Center, CHU Sainte-Justine, Montreal, QC H3T 1C5, Canada; 3Department of Anesthesiology, Université de Montréal, Montreal, QC H3T 1J4, Canada; 4Department of Nutrition, Université de Montréal, Montreal, QC H3T 1A8, Canada; 5School of Kinesiology and Physical Activity Sciences, Faculty of Medicine, Université de Montréal, Montreal, QC H3T 1J4, Canada; 6Department of Pediatrics, Université de Montréal, Montreal, QC H3T 1C5, Canada; 7Department of Pediatrics, CHU de Québec, Université Laval, Québec City, QC G1V 0A6, Canada; 8Research Center, CHU de Québec, Université Laval, Québec City, QC G1V 4G2, Canada; 9Department of Pediatrics, Université de Sherbrooke, Sherbrooke, QC J1N 3C6, Canada

**Keywords:** pediatric cancer, parental distress, problem-solving, dyadic coping, intervention development, intervention refinement, supportive care

## Abstract

A child’s cancer diagnosis profoundly impacts the psychological well-being of parents. To alleviate parental distress, researchers developed *Taking Back Control Together* (TBCT), a manualized six-session program targeting individual problem-solving skills and dyadic coping. The current study aimed to refine TBCT for future uptake across different sites. We invited potential interventionists and local stakeholders from three pediatric oncology centers (CHU Sainte-Justine, CHU de Sherbrooke, and CHU de Québec) to join the refinement team. The final working team comprised 26 professionals, including social workers, psychologists, researchers, coordinators, and parent-partners. The study included eight 50- to 90-min discussion sessions designed to stimulate conversation and facilitate the exchange of ideas and perspectives. We used framework analysis to identify and describe patterns within the qualitative data. The data were organized into three categories: (1) intervention description, which addresses changes in personnel, modes of delivery, and tailoring to accommodate different family structures; (2) content modifications, which include language simplification and visual enhancements; and (3) factors influencing TBCT’s future uptake, such as accessibility, participant satisfaction, clinician compensation, and flexibility in program delivery. The direct output of this research is a refined program with an updated manual, tools, and format adapted for use in different sites.

## 1. Introduction

A pediatric cancer diagnosis not only affects the child but also profoundly impacts the psychosocial well-being of the entire family [1]. Notably, parents often experience increased anxiety, uncertainty, depression, loss of control, and posttraumatic stress symptoms in response to their child’s cancer diagnosis [1,2,3,4,5]. Studies indicate that approximately 50% of parents of children with cancer report significant emotional distress throughout their child’s cancer trajectory [3]. A previous review reported that even five years post-diagnosis, parents may continue to experience uncertainty and anxiety [5]. These emotional and psychological responses often stem from the threat posed by cancer, the overwhelming amount of information given to caregivers about their child’s diagnosis, and the need for them to make critical treatment decisions [1]. The distress experienced by parents is not isolated, as it tends to extend into other spheres of family life. Established family roles and routines need to be restructured to accommodate treatment demands [1,2]. As a result, siblings of children with cancer may undertake more household responsibilities, restricting their involvement in extracurricular and social activities [6]. Couples may also face reduced intimacy and fewer shared activities [1,2]. Furthermore, studies demonstrate that a parent’s mental health is closely linked to their child’s emotional well-being and developmental outcomes, affecting areas such as school functioning or social skills [7,8,9]. Heightened parental distress may, in turn, make parents physically and emotionally unavailable, impacting the entire family unit [6]. Recognizing these cascading effects, it is therefore essential to provide psychosocial support for caregivers of children undergoing cancer treatment. To address this need, researchers have developed various interventions to alleviate parental distress and foster resilience within families affected by pediatric cancer [10].

Drawing on the strengths of previously recognized intervention programs (i.e., Bright IDEAS and Surviving Cancer Competently Intervention Program [10,11,12]), researchers have developed, refined, and tested the feasibility of *Taking Back Control Together* (TBCT) [2,13]. TBCT involves both individual and couple sessions, targets mothers and fathers alike, limits the number of sessions to lessen the burden on families, and incorporates a detailed manual with comprehensive instructions, role-playing exercises, and flexible guidelines for healthcare providers [2,13,14]. This structure allows caregivers to receive personalized and joint support to tackle the unique challenges their child’s diagnosis poses. TBCT aims to improve problem-solving skills and dyadic coping (i.e., a couple’s ability to deal with stress together [15]) among parents by empowering them to manage their emotions, fostering positive communication, and helping them regain a sense of control as they support their child with cancer. This intervention includes six sessions, four individual and two couple sessions lasting from 60 to 90 min.

In a feasibility study, TBCT demonstrated potential as a scalable and adaptable support mechanism for families navigating the complex challenges associated with pediatric cancer [13]. Yet, some important areas for improvement could be identified. In terms of reach, researchers obtained an enrollment ratio of 39% and a dropout ratio of 38%, well below the expected thresholds [13]. Several factors may have influenced the observed outcomes, including the possibility that eligible families were already receiving support, the prioritization of alternative interventions, disruptions caused by the COVID-19 pandemic, and parents’ perceptions that they did not need the psychosocial support offered by TBCT. Alternatively, barriers to reach may have resided in the program’s characteristics, such that sessions were administered by psychology interns and two postdoctoral fellows with clinical psychology degrees. As such, an additional collaborative relationship was required. Overall, the results demonstrated acceptable to excellent levels of treatment fidelity (77.3–84.3%). However, lower scores were observed in some areas, notably manual fidelity (64.2–75%), the use of worksheets and assignments (70%), and the program’s conclusion (73.3%) for the couple sessions [13]. These results suggest opportunities for improvement. Finally, the feasibility study revealed high ratings of acceptability (90%) and satisfaction (87%). Participants rated satisfaction more highly than relevance (82%), suggesting that, while they found the intervention satisfactory, there might be room for improvement in ensuring the program meets their specific needs. Given these results, a refinement study to optimize the TBCT program is essential. A refinement study would help identify core treatment components, refine the factors related to intervention delivery, and enhance overall effectiveness and efficiency (phase 1b, Obesity-Related Behavioural Intervention Trials—ORBIT—model) [16,17].

The present study aimed to refine the *Taking Back Control Together* intervention for future use in pediatric oncology centers across Québec. The objective is to adapt and extend the scope of TBCT across the Province of Québec to build an intervention capacity (phase 1b, ORBIT model) [16,17]. The specific objectives are (1) to optimize the program’s form and content to facilitate its adoption by pediatric oncology centers and families, (2) to adapt the intervention’s training and educational materials for caregivers and healthcare professionals to address the specificities and challenges of various pediatric cancer treatment centers, and (3) to identify, based on feedback and insights from involved professionals, the key factors that will likely influence the uptake of TBCT.

## 2. Materials and Methods

### 2.1. Team Description

The project’s initiators invited potential interventionists and key stakeholders across the Province of Québec to join the refinement team. We sought to assemble a diverse refinement team comprising individuals with key responsibilities in overseeing clinicians and physicians, affiliations to local treatment organizations, an interest in the program, potential for involvement as interventionists, and proficiency in French. Members from three pediatric oncology centers in university health centers (CHU) joined the research team: CHU Sainte-Justine, CHU de Sherbrooke, and CHU de Québec. These treatment centers collectively diagnose approximately 210 children per year. This diverse representation provides insights into province-wide challenges. Our final working team (N = 26) comprised various professionals, including six social workers (VS, JM, AS, NF, APT, RPR), one intern in social work (FM), four psychologists (DO, AR, MM, SS), two parent-partners (MR, ML), one doctoral student (AL), one psychology master’s student (NG), one research nurse (ÉM), seven research or clinical coordinators with a background in nursing or health sciences (ÉR, VB, IB, MÈL, ÉDS, JPB, MM), and three nursing and professional management professionals (MCC, SL, CC). The study included social workers and psychologists as potential interventionists to acknowledge the role variation across centers. In our environment, social workers typically consult with the entire family, whereas psychologists focus on the child, but they may also engage with parents in some cases.

The project received ethical approval from the Sainte-Justine Ethics Committee of the Sainte-Justine University Health Center (#MP-21-2023-5049). Team members involved in the refinement process served as co-researchers and are co-authors, contributing to and refining the intervention program. We report here on the process we collectively followed as a group. None of us were considered research participants.

### 2.2. Data Collection

Our refinement group operated through dynamic discussions and iterative decision-making processes facilitated by regular meetings. The present study included eight discussion groups, conducted by two psychologists (SS and DO), to facilitate the redesign process. Of these, five refinement meetings were held with stakeholders from all three pediatric oncology centers to refine the program collaboratively. Additionally, three pre-implementation meetings were conducted separately at each center, focusing on center-specific needs for implementing the redesigned program.

Discussion groups were chosen as the data collection method to stimulate conversations and facilitate the exchange of ideas and perspectives [18,19]. This approach supported the refinement process by allowing the group to respond to emerging challenges and providing rich data through participant interactions [20]. The research team selected discussion groups over individual interviews for their ability to elicit candid responses and enable participants to build on each other’s ideas, a phenomenon referred to as “piggybacking” [21]. This method was chosen to compare differences among pediatric oncology centers and enable more efficient data collection [22].

The meetings took place on the Microsoft Teams^®^ platform, lasted 50 to 90 min, and were recorded to document the team’s decisions. We transcribed the meetings via an online transcription platform (Sonix, 2024: https://sonix.ai/, San Francisco, CA, USA), followed by a manual review [23].

Meetings occurred from April 2023 to June 2024. The initial meeting introduced the intervention and the results of the previous feasibility study [13,14]. Each meeting outlined a specific agenda, and participants received homework to prepare for upcoming meetings. The structured discussion groups ensured an exploration of topics, fostering deeper understanding and collaborative problem-solving. Meeting dates and contents are summarized in Table 1.

### 2.3. Data Analysis

The present study used framework analysis to identify and describe patterns within qualitative data [24]. This method allows for organizing data into a matrix, with rows representing cases and columns reflecting key themes [24,25]. Its descriptive and structured nature makes it well-suited for health research that involves multidisciplinary teams [25]. The study aimed to fulfill prespecified information needs, thereby narrowing the analysis’ scope and enabling a structured and directed analysis [24]. We employed a deductive approach to guide the analysis, applying predetermined themes derived from a literature review and the discussion groups’ objectives. We intentionally opted for a deductive, rather than inductive, approach because the research focus was directed, and the analysis targeted specific research questions [26]. While a hybrid approach was initially considered, it was deemed unnecessary, as the predefined themes provided adequate guidance for the analysis (themes are detailed in the sections outlining the analyses for each objective). Two researchers (NG, SS) conducted multiple rounds of interpretation and discussion to ensure a thorough understanding and discussed the results with the team.

This analysis was computer-assisted with the QDA Miner 6^®^ software (Montreal, Québec, Canada), facilitating our analysis by tracking coded text [27]. The program allowed us to organize, annotate, and assign codes while minimizing errors and assisting in identifying recurring themes and patterns within the data.

The deductive (a priori) approach was grounded in themes derived from a literature review [24,25,28]. To explore our objectives, we analyzed verbal discourse to identify TBCT’s strengths and challenges, content modifications, and factors that may influence the intervention’s uptake.

For Objective 1, our analysis aimed to identify TBCT’s strengths and challenges using the Template for Intervention Description and Replication (TIDieR) checklist [29]. The TIDieR checklist, previously applied in health research to describe interventions, comprises twelve items covering an intervention’s rationale, materials, procedures, delivery, context, tailoring, and fidelity [29]. From these, we selected six key domains: (1) why, (2) what, (3) who provided, (4) how, (5) where, and (6) tailoring. To assess the rationale (*why*), we analyzed verbal cues reflecting stakeholders’ understanding of the intervention’s purpose and anticipated outcomes. For materials (*what*), we identified discussions about the program’s tools, focusing on whether they were viewed as helpful or hindering. To assess *who provided* TBCT, we identified exchanges on interventionists’ roles and responsibilities. Regarding *how* the intervention was delivered, we identified comments related to modes of delivery. Finally, to explore *tailoring*, we examined verbal cues indicating case-by-case adaptations.

For Objective 2, the study evaluated the program’s materials and tools by assessing verbal cues indicating content modifications to enhance clarity, relevance, and engagement. Stirman and colleagues (2013) identified twelve types of modifications, six of which were relevant to our analysis [30]. *Tailoring*, *tweaking*, or *refining* elements involves minor adjustments to the intervention’s materials and tools while maintaining their core principles. *Adding elements* refers to any inclusion of supplementary materials or tools that align with the intervention’s fundamentals. *Removing elements* identifies any materials excluded from the intervention package. *Substituting elements* involves replacing materials or tools with something different, such as replacing one tool with another. *Re-ordering elements* addresses activities completed in a different order from what was initially proposed. Finally, *repeating elements* refers to the repeated use of one or more tools.

For Objective 3, we selected elements from the Reach, Effectiveness, Adoption, Implementation, and Maintenance (RE-AIM) framework [31] and implementation outcomes [32,33]. Since group discussions focused on adapting TBCT and anticipating its implementation, our analysis centered on the program’s potential for *reach*, *maintenance/sustainability*, *perceived social validity* (*acceptability* and *appropriateness*), *costs*, and *fidelity*. To assess the intervention’s potential for *reach*—the proportion of the target population engaged in a program—we explored comments on location accessibility, outreach methods, and time commitment [34,35,36]. *Maintenance/sustainability* refers to the degree to which an intervention sustains over time [32,33,36]. To assess *maintenance/sustainability* potential, we identified discussions on integrating the program into routine practices and providing ongoing support for stakeholders, such as reusable resources and periodic check-ins. The analysis also focused on stakeholders’ perceptions to evaluate the program’s *perceived social validity*, encompassing *acceptability* and *appropriateness* [32,33]. Combining these notions under *perceived social validity* avoided redundancy and enabled a thorough examination of the social importance, acceptability, and significance of TBCT [37]. Here, we considered verbal cues that reflected stakeholders’ satisfaction with TBCT, as well as the intervention’s perceived fit and practicality. To assess *costs*, or the resources necessary for implementation, we identified discussions regarding clinician compensation and training [32,33]. Finally, *fidelity* refers to the delivery of an intervention as intended [32,33]. We categorized exchanges related to the quality of program delivery and adherence to guidelines or questions of standardization [32].

### 2.4. Data Analysis Quality

In line with Sargeant’s (2012) recommendations, we ensured qualitative research rigor and quality by focusing on (1) the data’s authenticity and (2) the analysis’ trustworthiness, which is akin to validity and reliability, respectively, in quantitative studies [38]. Data authenticity encompasses the quality of data and collection methods [38]. We achieved triangulation by including diverse perspectives from various pediatric oncology centers and roles, enriching the data’s depth. Meeting objectives were established after each discussion group. No predefined questions were prepared; instead, the project’s initiators outlined general topics, allowing for unbiased, open feedback. Stakeholder feedback was actively encouraged, which fostered an environment where diverse perspectives could be freely shared and considered. Discussions were also kept flexible to accommodate emerging needs. The analysis’ trustworthiness refers to the quality of the data analysis, which was achieved through transparent category development [38]. As such, meetings were transcribed and coded, focusing on a priori themes derived from existing models and literature, as well as the study’s objectives.

## 3. Results

Following the deductive classification of excerpts in Table 2, Table 3 and Table 4, we discuss the main results in the following sections. Essential statements are supported by excerpts referred to in parentheses and cited in the corresponding tables.

### 3.1. Aim 1: Strengths and Challenges of the Program

#### 3.1.1. Why and What

Team members clarified key points to ensure a shared understanding of the intervention’s purpose. For instance, they emphasized that the goal is not to resolve all pre-existing individual or couple problems but to help identify simple, well-defined problems that caregivers could later address with problem-solving tools (Table 2, citation 2.1). It was important to frame the intervention as a practical, supportive training rather than a therapy (citation 2.2).

The TBCT package includes three physical and informational documents: the intervention manual, the worksheets, and a self-help handout (“*Frequently Asked Questions*”). A parent-partner expressed concerns that some worksheets felt too evaluative, potentially hindering creativity (citation 2.3). Team members emphasized that the objective was not to evaluate families but to guide them. This feedback highlighted the need to make the tools more user-friendly than what was initially presented (cf. content modifications from Table 3).

#### 3.1.2. Who Provided

Discussions centered on the roles and qualifications of intervention providers. While the previous design trial relied on external interventionists (e.g., psychology postdocs or interns), this appeared to lead to coordination difficulties (citation 2.5). The decision was thus made to make the program available to experienced clinicians who would train parents in TBCT. This would reduce the need for basic instructions on establishing a relationship with parents (citations 2.4 and 2.6). The team also explored various role-related challenges, such as clinician-researcher role conflicts. One member noted the challenges in coordinating efforts between professionals, particularly social workers not involved in the project (citation 2.7). As a response, the refinement group emphasized the need for careful planning and communication between clinical and research teams. A group member noted that some clinicians perceive preventive interventions as outside their responsibilities, which could impede adoption, especially if it increases their workload. A proposed solution was to involve interns supervised by clinicians to deliver the intervention and distribute the workload (citation 2.8). Additionally, the group had differing views regarding who should provide the intervention. Two patterns emerged: some favored continuity, with the same clinician conducting both the initial clinical assessment and TBCT sessions (citations 2.9 and 2.12), while others preferred flexibility by involving clinical psychology interns specifically for TBCT (citation 2.10) or transferring families after their initial assessment to other professionals who would then train parents in TBCT (citation 2.11).

#### 3.1.3. How and Where

The group discussed the intervention’s delivery modes. Team members expressed concerns about whether parents would take the time to write their answers by hand, noting that people are increasingly accustomed to using electronic devices (citation 2.14). Therefore, to accommodate parents’ preferences and facilitate adoption, clinicians agreed to offer electronic and paper-pencil options if required (citation 2.13). Yet, team members also emphasized that having tangible, paper-based tools would help parents externalize their problems. Providers noted that they could envision themselves using the worksheets as collaborative tools (citation 2.15). Additionally, to address scheduling challenges and logistical barriers, the clinicians agreed to offer the intervention face-to-face or by telehealth (citation 2.16).

#### 3.1.4. Tailoring

The refinement team emphasized the importance of allowing flexibility in intervention intensity, noting that some families may master problem-solving skills quickly, allowing some sessions to be skipped (citation 2.17). The group noted that problem-solving training could benefit more socially vulnerable parents, as socially secure families may already possess or quickly acquire these skills (citation 2.18). Moreover, the research team discussed tailoring session content to accommodate various family configurations. For single-parent families, only the four individual sessions will occur (citation 2.19). For separated parents, providers agreed to prioritize parents who share parenting responsibilities (citation 2.20). Hence, fostering a co-parenting dynamic and emphasizing collective problem-solving might be more appropriate than focusing on communication and intimacy issues (citation 2.21). For blended families, the group discussed including the child’s stepparent. Clinicians also recognized TBCT’s limitations for couples with long-standing issues or conflicts; in such cases, they agreed to focus only on individual sessions.

**Table 2 curroncol-32-00253-t002:** TBCT’s strengths and challenges mentioned by the refinement team across key items from the Template for Intervention Description and Replication (TIDieR) checklist.

Themes *	Strengths (+)	Challenges (−)
Why	2.1 Social worker (Site 1): *From what I understand from the proposed meetings*, *it’s not that we want to solve all the marital problems that existed before or that exist now*, *but that they* [caregivers] *become aware of them and develop certain tools.*	
	2.2 Psychologist (Site 1): *We were also able to present to parents the fact that we weren’t there to provide psychotherapy*, *but more to equip them*, *in fact*, *to give them complementary resources to help their child.*	
What		2.3 Parent-partner (Site 1): *I remember when I filled out this type of grid maybe two years ago*, *I found that sometimes it made you feel evaluated*, *and then you have this desire to give a good answer when the process resembles an exam. Do you follow me? So*, *there’s a feeling of having to give the correct answers and not being in a reflective*, *brainstorming*, *or creative mode.*
Who	2.4 Psychologist (Site 1): *For this program*, *it’s going to be completely different because we’ll be working with clinical practitioners.*	2.5 Psychologist (Site 1): *The main barriers we faced were mostly logistical. This included the lack of time from the parents and scheduling conflicts with clinical activities. What happened was that we had external people coming in to lead the activities.*
	2.6 Psychologist (Site 1): *What happens is that*, *since we’re dealing with people who are already in contact with the parents*, *we can explain less.*	
		2.7 Social worker (Site 2): *My situation is a bit different because there is a social worker in oncology who isn’t involved in the project. So*, *for me*, *it’s more a matter of coordinating the two groups to see what we can do*, *how we can deal with this.*
		2.8 Psychologist (Site 2): *So*, *I get the impression that this kind of program*, *or preventive intervention isn’t part of* [my colleagues’] *job. That’s why maybe it’s the student interns who will do it or me.*
	2.9 Social worker (Site 1): *We choose the families when we accept a request. In other words*, *we receive the requests and share them among ourselves in a balanced way.* Social worker (Site 1): *Depending on availability. We have a table that allows us to group all the requests and then separate them.*	
	2.10 Psychologist (Site 2): *I’m definitely taking advantage of the PhD student I’ll have in September*, *who will also be part of this project. It’s already in the works. It’s already been discussed with the students who want to come and do their practicum here.*	
	2.11 Social worker (Site 2): *We systematically get involved. For the families we are already working with*, *depending on when the program starts*, *it will either be us or the intern who will be the most suitable. If these are families where we are already starting to reduce our follow-up intensity*, *we may pass the torch to the intern.*	
	2.12 Psychologist (Site 3): *Families would probably already have an interventionist assigned to the case. I think we will want to maintain that continuity.*	
How	2.13 Social worker (Site 2): *I think it would be better for some families to use paper and pencil because they might not be comfortable using the Internet for such tasks. For others*, *electronic methods would be easier.*	2.14 Research nurse (Site 3): *Well*, *I wonder if parents will actually take the time to write. You know*, *people are getting more and more used to using their electronic devices to fill out forms or answer questions.*
	2.15 Social worker (Site 1): [Providing paper-based tools] *is a good idea because it provides a tool to help take the thoughts out of the parents’ heads. You know*, *just talking about it can become confusing. Getting things out and seeing them externally is a very good idea.*	
Where	2.16 Psychologist (Site 1): *Offering* [the intervention] *remotely*, *like through videoconference*, *allows us to offer* [the intervention] *at convenient times*, *not at the hospital—sometimes in the evening or when the parents are back home*, *so they’re no longer at the hospital.*	
Tailoring	2.17 Psychologist (Site 1): *For me*, *it happened that in four sessions*, *we addressed four different problems because some people were very efficient. But it also happened that in four sessions*, *we only worked on two problems. So*, *it really depends on the parent’s pace.*	
	2.18 Psychologist (Site 1): *Following the problem-solving steps is particularly effective for families that are less educated.*	
	2.19 Psychologist (Site 1): *There is also the case of single-parent families*, *where we will only meet with one parent for the first four sessions.*	
	2.20 Psychologist (Site 1): *But it’s clear that* [for separated parents] *we don’t address topics such as marital intimacy.* Psychologist (Site 2): *Would it be more of a co-parenting intervention?*	
	2.21 Psychologist (Site 1): *We didn’t do the couple session in the same way because working on conjugal intimacy with separated parents is more complicated. But we did work on the first part*, *actually problem-solving*, *because there are custody issues and things like that.*	

* Themes were derived from the Template for Intervention Description and Replication (TIDieR) checklist [29]. DOI: 10.1136/bmj.g1687. Site 1: CHU Sainte-Justine; Site 2: CHU de Québec; Site 3: CHU de Sherbrooke.

### 3.2. Aim 2: TBCT’s Materials and Tools

#### 3.2.1. Refining Materials and Tools

Tailoring/tweaking/refining content emerged as a significant theme, involving minor adjustments to TBCT’s materials. Team members suggested consolidating the separate manuals for parents and interventionists to avoid confusion and redundancy. Discussions also emphasized correcting typos, errors, and inconsistencies (Table 3, citation 3.1). The team recommended modernizing the manual’s visuals for a more engaging look (citation 3.2). Following these remarks, the refinement group agreed to rectify these errors when creating the unified manual. Moreover, the group explored simplifying the worksheet titled “Evaluation of Solutions”. The revised worksheet features two main sections, “Advantages” and “Disadvantages”, making it easier to understand and complete (citation 3.3). Participants suggested revising the language in the presented cases to accurately represent families in pediatric oncology and avoid stereotypical portrayals. As such, we modified “*Marie’s Story*” to indicate that she was on disability leave or caregiver benefits rather than a stay-at-home mother (citation 3.4). Finally, phrases such as *“It’s as simple as that”* might not resonate well with all families and might be perceived as simplifying their struggles (citation 3.5). Thus, the group switched from *“simple”* to *“practical”.*

#### 3.2.2. Adding or Removing Materials and Tools

The refinement group identified opportunities to enrich existing tools by adding elements. Team members proposed including explanatory content to clarify ambiguous materials, using simpler language (citation 3.6). The team included interactive elements within the manual and worksheets, such as *“Interesting fact!”* and *“Tips and tricks*”, as well as illustrations (citation 3.7). Additionally, the group suggested adding summary pages at the end of the manual to outline problem-solving steps, serving as a reference guide (citation 3.8). To promote user convenience, the team added QR codes and hyperlinks to direct users to support material. The research team also identified several elements in need of revision. Group members agreed that the manuals contained too many verbatims, often too detailed for experienced clinicians (citations 3.9 and 3.10). They proposed reducing these verbatims to make the refined manual more concise and relevant. As for the worksheets, some were deemed repetitive, such as the “Imagined Solutions” worksheet (citation 3.11). Participants also identified the “Anticipate Risky Situations” worksheet as redundant and confusing, proposing to eliminate it and integrate its content as a verbatim.

#### 3.2.3. Substituting, Re-Ordering, or Repeating Materials and Tools

Team members identified opportunities to replace materials with better alternatives. As such, they proposed prioritizing the “*Canoe Trip*” video instead of its written version, a short story illustrating problem-solving steps (citation 3.12). The group recommended reordering activities to improve the program’s flow. For instance, the research team integrated activities initially planned as homework within face-to-face sessions, such as watching the “*Canoe Trip*” (citation 3.13). The group suggested assigning “*Marie’s Story*” as homework after Session #1 and decided that Session #2 would begin with a discussion on the reading, bridging the sessions and allowing clinicians to summarize the case if needed (citation 3.14). The refinement group proposed distributing the “*Frequently Asked Questions*” handout at the end of Session #4 for caregivers to review with their partners before the couple sessions. The team suggested repeating specific elements, such as having parents rewatch the “*Canoe Trip*” as homework individually or with family members (citations 3.15 and 3.16). The team viewed this dual approach of in-session viewing and at-home review as beneficial for comprehension and collaborative learning.

**Table 3 curroncol-32-00253-t003:** Remarks and suggestions regarding the materials and tools used in TBCT.

Themes *	Codes	Examples of Verbal Statements
Tailoring/ tweaking/ refining elements	Simplification and clarification (manual revisions and error corrections)	3.1 Psychologist (Site 2): *Some things were difficult to connect between the title of the worksheet and the worksheet next to it. We didn’t have the same titles. I have the impression that… I don’t know if you can tell me because sometimes it’s the…Or the number or the worksheet didn’t have the same title.*
	Visual enhancements	3.2 Social worker (Site 1): *I don’t want to seem like I’m*, *like*, *quibbling. Or maybe it’s not modifiable*, *but I thought that for the “Virtuous Circle” and “Managing Stress Together,*” *I believe there are three of them—the “Think Feel Connection” … I think the visuals are a bit outdated*, *I think it’s a bit ‘90s [1990s].*
	Layout	3.3 Psychologist (Site 1): *Yeah*, *in fact*, *you’ve got four; there are four domains on which each solution is evaluated: emotional well-being*, *commitment*, *short- and long-term costs*, *as well as short- and long-term benefits. I think…* Social worker (Site 1): *Could it be separated into advantages and disadvantages and then put in brackets…* Social worker (Site 1): *…emotional well-being*, *commitment*, *time/effort*, *costs.* Psychologist (Site 1): *Yeah*, *they’re examples of domains that*, *I think*, *are good*, *but we should put them separately. We could put two paragraphs*, *two sections*, *and then examples of domains in brackets. That would be great!*
	Cultural and contextual adaptations (case study adjustments and inclusive language)	3.4 Social worker (Site 1): *There’s a subtle detail in “Marie’s Story”: we assume she doesn’t work. It seems that today*, *knowing that most mothers work*, *there’s an issue. Would it have been possible to make a slight change and*, *I don’t know*, *say that she’s on disability leave? Because it’s as if she’s a stay-at-home mom*, *and I thought that was a bit stereotypical.* Social worker (Site 2): *For me*, *she was on caregiver benefits.*
		3.5 Research or clinical coordinator with a background in nursing or health sciences (Site 1): *“Identifying the problem. It’s as simple as that.” Well*, *maybe it won’t be simple for the family. It just made me think. I thought “My God*, *if I try to find…” I haven’t been in that situation*, *but I guess… sometimes you can kind of get overwhelmed when trying to find a problem. Well*, *I don’t know if it’s that simple.* Research nurse (Site 3): *I’m thinking about “a practical method.” You know*, *it’s practical*, *it’s practical…*
Adding elements	Explanatory materials and clarification	3.6 Psychologist (Site 1): *So*, *the verbatim to add is to explain the figure*, *in fact*, *on stress and the impact on communication…in simpler words*, *in fact.*
	Interactive elements	3.7 Psychologist (Site 1): *And all these little messages of support*, *I find that really interesting because in the previous manual*, *we didn’t have that. It was very*, *very structured.* Psychologist (Site 1): *The messages or drawings also on the first one with the mountain*, *I get the impression it’s a rower*, *a lighthouse. It’s*, *it’s interesting*, *I think.*
	Clarification	3.8 Social worker (Site 1): *Perhaps to have a little summary*, *really a concise reminder of the steps.*
Removing elements	Simplification	3.9 Psychologist (Site 1): *There are a lot of verbatims*, *and then there are verbatims that are too simple for you because you’re already clinicians. So*, *I think we need fewer verbatims.*
		3.10 Psychologist (Site 1): *There are verbatim examples to use with parents*, *but we’ve removed all that for the upcoming activity*, *because we’re dealing with people who already know how to interact with parents.*
	Redundancy	3.11 Social worker (Site 2): *I had a question about Worksheet E “Imagined Solution” because*, *in my head*, *in “Evaluating Solutions,” you know*, *we’re already thinking about what the impact is going to be*, *what it’s going to be… I didn’t understand the purpose of “Imagined Solution” because*, *in my head*, *we’d already done it in Worksheet D? To try and see what the result was going to be and so on.*
Substituting elements	Clarification	3.12 Social worker (Site 1): *The “Canoe Trip.” I seem to have seen it somewhere in the manual to read the story*, *but we agree that it’s been replaced by this video? The “Canoe Trip” video.*
Re-ordering elements	Session flow	3.13 Social worker (Site 2): *In my mind*, *we watched the video at the beginning of the meeting*, *after presenting the program. I didn’t see it as a homework assignment in my head.*
		3.14 Social worker (Site 1): *Could we think about doing*, *you know*, *we*, *we give that* [*“Marie’s Story”*] *as a reading*, *but we still prepare a summary. So*, *at the second meeting*, *we could start with, “What did you think of the reading?” If that’s been done*, *we can go and get some feedback. Then*, *if they haven’t done so*, *we could give them a summary so that they can do it for the next session.*
Repeating elements	Practical flow of sessions	3.15 Psychologist (Site 1): *Regarding your comment about the “Canoe Trip,*” *many people have actually watched the “Canoe Trip” with their children.*
		3.16 Psychologist (Site 1): *Well*, *at the very least*, *it could also be a homework assignment. Something we’d advise them to do at the end would be to review the “Canoe Trip.” Maybe it’s easier to watch a video than to feel like you’re studying papers.*

* Themes were derived from content modifications proposed by Stirman and colleagues (2013) [30]. DOI: 10.1136/bmj.g1687. Site 1: CHU Sainte-Justine; Site 2: CHU de Québec; Site 3: CHU de Sherbrooke.

### 3.3. Aim 3: Factors Likely to Influence Future Uptake

#### 3.3.1. Reach

Reach refers to the proportion of the target population engaged in a program. Parents often struggle to manage childcare while attending hospital appointments with their sick children (Table 4, citation 4.1). Thus, the team recognized that arranging childcare could pose a barrier to participation in TBCT. Certain medical conditions also make it inappropriate for families to participate in support interventions, such as advanced cancer, i.e., a condition that cannot be cured with standard treatment (citation 4.2). Recruiting families from distant locations adds another layer of complexity, as long travel times can hinder engagement (citation 4.3). To address these barriers, the refinement team decided to offer remote sessions through telehealth technologies, allowing at-home participation and reducing travel and childcare needs (citation 4.4). Coordinating with medical staff and ensuring the program aligns with clinical activities may also improve accessibility (citation 4.5). The process involves obtaining medical and psychosocial clearance from doctors, pivot nurses, social workers, and psychologists before approaching families, ensuring the program is presented at the right time (citation 4.6). As such, the group decided that recruiting families 4 and 16 weeks post-diagnosis would be appropriate. Since families may be spread across remote locations, team members emphasized the importance of strategically introducing the intervention (e.g., during hospital stays) to maximize participation (citation 4.7). Clinicians from one site noted additional barriers, such as the need to balance their responsibilities across several departments, possibly complicating the uptake of TBCT (citation 4.8).

#### 3.3.2. Maintenance/Sustainability

The group mentioned that integrating the program into routine practices could foster its long-term use. A psychologist involved in the previous trial noted that he still uses TBCT tools in his current practice (citation 4.14). Team members mentioned that some clinicians in the current project pre-tested the tools in their daily clinical work (citation 4.12). However, one member raised concerns about integrating the intervention into her routine practice, as her approach is more open and interactional (citation 4.13). This highlights the importance of balancing structure and flexibility. As a result, the refinement team emphasized the value of follow-up meetings with interventionists to discuss obstacles and share solutions (citation 4.9). The group also discussed using detachable worksheets for parents to define problems and work on solutions, promoting sustained use after formal sessions (citation 4.10). A parent-partner noted the importance of providing a reusable guide for families, instilling a desire to continue the process (citation 4.11). A team member mentioned that while many parents were familiar with the intervention strategies, the program’s structured approach would help them organize their efforts more effectively (citation 4.14).

#### 3.3.3. Perceived Social Validity

Perceived social validity assesses how stakeholders view the intervention’s importance, acceptability, and significance. Team members expressed satisfaction with the program (citation 4.15) and noted improvements in the refined manual compared with earlier versions (citation 4.16). The research team found the intervention interesting, well-structured, simple, and applicable (citation 4.19). The group highlighted the relevance of videos and other tools, noting that they simplified complex contexts and made them more relatable for parents (citation 4.20). Team members also underscored the program’s complementarity with existing services, noting that TBCT is a quick and intensive intervention that complements, rather than replaces, existing practices (citation 4.21). This should facilitate smoother integration into providers’ workflows without significantly adding to their workload (citation 4.22). The group also reported that families valued the psychosocial component of the previous trial (citation 4.17, see the following reference for more information on the overarching initiative [39]). A notable endorsement came from a parent-partner who, despite not participating in the prior study, expressed that such tools would have helped her and her family during treatment (citation 4.18).

#### 3.3.4. Costs and Fidelity

Costs assess the resources required for program uptake. Team members discussed funding to compensate professionals for participating in meetings and training activities. In the present refinement activity, compensation came from a research grant budget (citation 4.23). The group also emphasized the need to train clinicians in intervention delivery (citation 4.24). This should ensure that the providers across all sites are well-prepared and receive consistent instruction. Training requires resources to cover the preparation and delivery of training sessions, as well as compensation for clinicians’ time spent in training. Since the data were collected for the present report, we organized three structured training sessions.

Fidelity refers to delivering a program as intended. The refinement group discussed the need for flexibility, allowing practitioners to adapt the intervention to their style and the needs of families. While the manual provides verbatims as guides, providers insisted they could adapt them to their language and style (citation 4.25). Team members emphasized developing a version of the intervention that, while adaptable, can be harmonized across sites to ensure consistency and maintain TBCT’s integrity (citation 4.26). Finally, to collect evidence on the resources used by participating professionals, the group agreed to use brief checklists to track their activities during each session, including session content, materials, tools, and homework (citation 4.27).

Here is a link to the materials from the refined intervention, which are publicly available in Canadian French and English: https://doi.org/10.17605/OSF.IO/KVHUF [40].

**Table 4 curroncol-32-00253-t004:** Factors likely to influence TBCT’s future uptake are categorized by the Reach, Effectiveness, Adoption, Implementation, and Maintenance (RE-AIM) framework domains and implementation outcomes.

Themes *	Codes	Barriers to Implementation	Facilitators to Implementation
Reach	Location accessibility	4.1 Psychologist (Site 1): *For example*, *I remember a woman who had several children*, *and her problem was how to keep her children at home when she had to come to the hospital with her child and stay there because they had come from far away.*	
		4.2 Psychologist (Site 1): *There are medical conditions for which this lifestyle intervention is not possible.*	
		4.3 Psychologist (Site 2): *The parents come from Eastern Québec*, *so we don’t always have access to parents in the Québec City area. As a result*, *we might have virtual meetings.*	
			4.4 Psychologist (Site 1): *We also mentioned the question of telehealth options*, *such as doing it when the child is at home when they can be in their room playing to avoid this childcare problem.*
	Outreach methods		4.5 Research or clinical coordinator with a background in nursing or health sciences (Site 1): *One of the inclusion criteria for the project includes prior approval by the medical team. This means that if we tell you* [the clinicians] *we want to approach a particular family for the project and you decide that “Oh no,*” *there are major problems or specific issues with the information that you have*, *and you decide*, *based on your experience and clinical knowledge*, *that it wouldn’t work*, *that it doesn’t apply to that family for various reasons*, *you also have an opinion on this before we recruit a participant.*
			4.6 Research or clinical coordinator with a background in nursing or health sciences (Site 1): *Once I have the doctor’s authorization*, *I also check with the nurse and all to see when the best time is to approach them to present the project. Once the patient has been included in the study*, *the various clinicians are informed that they have been recruited.*
			4.7 Research or clinical coordinator with a background in nursing or health sciences (Site 2): *The ideal would be to catch patients while they’re hospitalized because many families live far away from the hospital.*
	Time commitment	4.8 Social worker (Site 3): *We cover all the other departments. So*, *pediatrics*, *neonatology*, *maternity*, *intensive care*, *and high-risk pregnancies.*	
Maintenance/ Sustainability	Sustained use		4.9 Psychologist (Site 2): *I think it could be interesting to follow up on the meetings.*
			4.10 Psychologist (Site 1): *We also have detachable worksheets that can be given to parents to work on the different objectives*, *the operationalization of solutions and objectives*, *and so on.*
	Guidance and empowerment		4.11 Parent-partner (Site 1): *It’s about giving the person a guide so they can use the same process again later and then making them want to do it too.*
	Integration into routine practices		4.12 Psychologist (Site 1): *When I introduced the tools*, *some of* [the clinicians] *already started using them. They wanted to pre-test* [the tools], *and it’s actually working well.*
		4.13 Social worker (Site 1): *I wonder how I’m going to integrate* [the intervention], *as I often improvise things.*	
			4.14 Psychologist (Site 1): *When I present* [the intervention]*—and again*, *I still use* [the intervention] *in my practice today with my patients—these are things that patients are already doing. And even when we did this with the parents*, *afterwards*, *the parents would say*, *“Well*, *yes*, *but I’ve always done that. But now I have a structured method and terms that I can use.”*
Perceived social validity	Satisfaction		4.15 Social worker (Site 1): *I find* [the intervention] *very interesting*, *and I’m excited about it.*
			4.16 Research or clinical coordinator with a background in nursing or health sciences (Site 1): *I really like the presentation. It’s a big improvement compared to the manuals we had in VIE 1.0* [Valorization, Implication, and Education (VIE): the overarching initiative under which TBCT was developed].
			4.17 Psychologist (Site 1): *We realized that VIE was also greatly appreciated by the participants and their families*, *especially the psychosocial component.*
			4.18 Parent-partner (Site 1): *I find it really*, *really interesting because I didn’t have that kind of support. After reading about it*, *I realized that it’s the tool that*, *looking back*, *would have helped when I think about what was missing in terms of support.*
	Perceived fit		4.19 Psychologist (Site 3): [The intervention] *is perfectly applicable. It’s quite structured. Simple too.*
			4.20 Social worker (Site 1): [The *“Canoe Trip”*] *is very*, *very simple to understand. It’s something we can all relate to; we can all think of a similar situation where we don’t have control*, *like with an illness. It’s a great way to simplify things and provide a clear image. Then*, *we can reuse it at various stages. Then*, *the “Parents’ Testimony” adds a more specific application to their situation. We start with a simple*, *metaphorical image and then include a testimony that speaks directly to their reality*, *making them feel understood. Even though we often normalize things for the parents*, *our role is to reflect the experiences of other parents but hearing it directly from them adds an extra touch.*
	Practicality		4.21 Psychologist (Site 1): *So*, *it’s not an additional intervention; it’s something that enriches or complements what we usually do with parents. In other words*, *it may seem like it takes a lot of hours because we need to see the families multiple times*, *but at the same time*, *our intervention is intensive and ends quickly.*
			4.22 Nursing and professional management professionals (Site 3): *So*, *regarding social work*, *it’s true that* [the intervention] *doesn’t require a high level of effort; it won’t necessarily increase the workload.*
Costs	Clinician compensation		4.23 Research or clinical coordinator with a background in nursing or health sciences (Site 1): *But I want to mention that we also have a budget to pay professionals. It’s important to point this out. Whether it’s for training or attending meetings*, *we have funds allocated to the centers*, *which should be used to pay the professionals as well.*
	Clinician training		4.24 Research or clinical coordinator with a background in nursing or health sciences (Site 1): *We talked about training provided to professionals.*
Fidelity	Flexibility		4.25 Psychologist (Site 1): *The written instructions are just guidelines. They suggest what to say if ever you get stuck or need help. But ideally*, *for this type of simple intervention*, *it’s best to follow its spirit but put it in your own words.*
			4.26 Psychologist (Site 3): *We can be flexible*, *but the idea is to achieve consistency so that the different centers offer the same services as much as possible.*
	Adherence		4.27 Research or clinical coordinator with a background in nursing or health sciences (Site 1): *It’s a small checklist for the practitioners because the goal is to see if it’s applicable everywhere*, *in all settings. Are there tools that we ultimately aren’t using?*

* Themes were derived from the RE-AIM framework and implementation outcomes [31,32,33]. DOI: 10.1136/bmj.g1687. Site 1: CHU Sainte-Justine; Site 2: CHU de Québec; Site 3: CHU de Sherbrooke.

## 4. Discussion

The present study aimed to refine *Taking Back Control Together*, a manualized intervention designed to support parents confronted with pediatric cancer (phase 1b, ORBIT model) [13,14,16]. Refinement was needed to facilitate the intervention’s future uptake in three pediatric oncology centers across the Province of Québec. Our results highlight the importance of refining interventions to cater to the needs of end-users. In terms of intervention description, our refinement team discussed the intervention’s purpose, changes in personnel, delivery modes, and accommodations for diverse family structures. Our refinement team made numerous content modifications based on collaborative exchanges. We created a unified manual, corrected errors, improved visuals, and revised the language to ensure cultural and contextual relevance. We also added explanatory content and interactive components, reduced verbatim text, eliminated redundant tools, replaced a reading assignment with its video version, integrated homework activities within face-to-face sessions, reorganized the sequence of other tools, and repeated key elements. Additionally, the refinement team identified several factors influencing TBCT’s future uptake, including location accessibility, outreach methods, time commitment, sustained use, guidance and empowerment, integration into routine practices, satisfaction, perceived fit, practicality, clinician compensation and training, flexibility, and adherence.

### 4.1. Drawing Lessons from Feasibility Data

One key finding centered on the roles and qualifications of interventionists. The previous feasibility study suggested that relying on psychology interns and postdoctoral fellows may have contributed to lower fidelity scores [13]. Gearing and colleagues (2011) identify training as a core component influencing treatment fidelity, encompassing elements such as skill level, education, experience, and implementation styles [41]. To address this, TBCT is now delivered by experienced clinicians, such as social workers and psychologists specializing in pediatric oncology. We hope this shift will reduce coordination challenges and improve treatment fidelity. To ensure consistent delivery, the project’s initiators organized three structured training sessions, one at each participating site. Moreover, monitoring intervention delivery, another key component impacting fidelity, includes assessing competence, such as clinicians’ ability to engage with participants [41]. By engaging experienced clinicians in the delivery of TBCT, we reduce the need for basic instructions on building relationships with parents.

The disparity between satisfaction and relevance ratings observed in the preceding feasibility study underscores the importance of ensuring the intervention aligns with the specific needs of parents. The current study involved two parent-partners in the refinement process: one who had previously participated in TBCT and one who had not. This dual perspective provided valuable information about the program’s strengths and areas for improvement. For instance, the parent-partner who participated in TBCT noted that some of the worksheets’ structured format felt evaluative, akin to an exam. This created a sense of needing to provide “correct” answers rather than fostering reflective, brainstorming, or creative thinking. In response, the research team revised the program’s format and adapted the training materials to minimize this evaluative tone. Additionally, the parent-partner who had not taken part in the previous trial was debriefed on the refinement process and presented with the updated materials. They shared that the tools addressed a gap in the support they previously received throughout their child’s treatment. Such feedback suggests that the refined intervention may better meet the diverse needs of parents navigating pediatric oncology care. Increasing the program’s relevance could, in turn, help shift parents’ perceptions that they do not need the psychosocial support offered by TBCT, a barrier identified in the previous trial [13]. These insights not only validate the refinements made but also highlight the potential of the refined intervention to address critical gaps in support.

The reach variables in the previous feasibility study did not meet the anticipated benchmarks, with low enrollment and high dropout rates attributed to several factors, including disruptions caused by the COVID-19 pandemic [13]. Our refinement team identified additional barriers to reach, such as childcare challenges and long travel distances for families in remote areas. The team proposed different strategies to navigate these barriers and improve accessibility, namely the use of telehealth technologies to deliver sessions remotely. Before the COVID-19 pandemic, telehealth adoption faced various challenges, such as difficulties with technology use, limited internet access in rural or underserved regions, inadequate reimbursement for services, location restrictions, and privacy regulations [42,43]. Nonetheless, the pandemic accelerated the shift to telehealth due to frequent lockdowns, work-from-home mandates, and the need for safe patient care. Researchers have reported that remote sessions minimize travel times, allowing individuals to access services from the comfort of their homes [42,44,45,46]. Additionally, telehealth reduces the need for childcare arrangements, addressing a key barrier to participation in psychosocial interventions identified in our study. Thus, remote sessions represent a promising option to improve reach in future TBCT trials.

### 4.2. Strengths and Limitations

A notable strength is the study’s collaborative and iterative refinement process, which engaged diverse stakeholders from three pediatric treatment centers, including potential interventionists and parent-partners. Stakeholders offer unique perspectives rooted in direct knowledge and experience [47]. Stakeholder engagement can provide information that is more relevant and responsive to end-user needs. For our refinement study, this approach ensured that TBCT was tailored to real-world clinical settings and caregiver needs. Nonetheless, we should acknowledge the present study’s limitations. First, our study engaged stakeholders from three French-speaking pediatric treatment centers in Québec, a culturally and linguistically specific province of Canada. This context limits the generalizability of our findings, as the refinements made to TBCT may not fully align with the needs or preferences of other cultural contexts. To address this, we adapted and translated the intervention into English to ensure accessibility. Future research should test these refinements in different settings, such as English-speaking provinces in Canada. Second, while structured and goal-oriented, the deductive analytical approach may have constrained the exploration of emergent themes in the data. That is, the deductive approach restricts the identification of responses to pre-determined domains. Third, some suggestions and feedback from our refinement group were contradictory. After reformulating certain elements for clarification, discrepancies remained, requiring us to adopt the majority consensus. For instance, while some team members expressed a preference for paper-based tools, others advocated for electronic formats. To balance these differing perspectives, we decided to offer both electronic and paper-pencil options to promote accessibility and accommodate varying user preferences. Fourth, despite our efforts to integrate the group’s recommendations, our capacity to implement certain suggestions was constrained by available resources (e.g., time and funding). For example, while some team members recommended developing a mobile app, financial and technical resource constraints prevented us from doing so. In addition to initial development costs, maintaining an app would require ongoing updates to ensure compatibility with different operating systems and address potential bugs—demands that exceeded our available resources. Therefore, we made the materials available online through web links to address this suggestion. Additionally, while our refinement team discussed the importance of cultural representation in video content, we did not have the means to produce new videos. We also recognize the need to represent sexually diverse couples and to consider how socio-economic barriers and varying levels of literacy may influence parents’ experiences in navigating their child’s cancer journey. One potential avenue for addressing this limitation could involve leveraging AI-generated video or audio scenarios that reflect diverse backgrounds while maintaining cost efficiency.

## 5. Conclusions

The current study documented the refinement of TBCT, an intervention designed to reduce the emotional distress of parents of children with cancer by targeting problem-solving skills and dyadic coping. The direct output of this research is a refined program with an updated manual, tools, and format adapted for the Province of Québec, which we expect to improve feasibility outcomes (materials are available in Canadian French and English: https://doi.org/10.17605/OSF.IO/KVHUF [40]). Our work serves as a necessary intermediate step between an initial feasibility study and capacity building in several sites and should help conduct future pilot studies. By engaging stakeholders from three pediatric oncology centers during this early phase, we collected important feedback that should help the future use of this supportive program in pediatric cancer care settings.

## Figures and Tables

**Table 1 curroncol-32-00253-t001:** Outline of the meeting dates, agenda, discussions, and homework of the refinement team of the intervention *Taking Back Control Together* (TBCT).

Meeting	Date	Agenda and Activities	Key Discussions	Homework
1	5 April 2023	(1) Introduce TBCT’s components and tools; (2) Discuss collaborative goals: focus on adapting the program to assess its feasibility for broader application; and (3) Review the objectives and content of educational materials.	Presentation of TBCT within the broader research environment. Topics discussed: motivational support, project timeline, maintaining uniformity across centers, involvement of psychology students, adapting the intervention for separated parents, defining professional roles, and next steps for refinement.	Professionals are to watch two videos (“*Canoe Trip*,” a short, animated video used as a metaphor to illustrate problem-solving steps, and “*Parents’ Testimony*,” a video featuring real parents sharing their experiences with their child’s diagnosis) and read selected pages of the interventionist’s manual.
2	19 April 2023	(1) Present the structure and activities of the six sessions; and (2) discuss the relevance of tools: explore potential adaptations and emerging needs.	Topics discussed: feedback on tools (discrepancies, flexibility in use, and suggested improvements), implementation feasibility (workload integration and timing challenges), video tools (positive feedback and challenges in creating new videos), and proposed solutions (digital adaptations).	Professionals are to read the problem-solving case (“*Marie’s Story*,” a written story that thoroughly demonstrates the six problem-solving steps taught in the intervention, serving as a concrete example to help parents understand and apply the method).
3	10 May 2023	(1) Review TBCT’s procedures, structure, barriers and facilitators, and discuss practical role-playing scenarios; (2) Discuss psychosocial protocol: evaluate the intervention’s format and content and propose potential adaptations, focusing on feasibility; and (3) Discuss pre-post assessment tools and data collection methods.	Topics discussed: structure and flow of each session, simplifying the manual and worksheets, reorganizing the presentation of tools and homework, and tailoring the intervention for different family configurations (couple sessions to be modified in some contexts).	A draft intervention program is circulated (format and content of the intervention for each session in table form), along with the associated tools. Professionals are to review it and suggest revisions if necessary.
4	21 June 2023	Validate TBCT’s content and format.	Review of refinement process and feedback from parent-partner who participated in a previous trial. Topics discussed: need to enhance visuals, clarity, and organization of the revised manual, delivery modes, distribution of materials, using checklists to document activities, training sessions, and strategies for future implementation.	Following this meeting, the intervention program is validated (format and content). - Simplified intervention guide (table per session, verbatim, and essential tools) - Parents’ manual - Tools per session Following these four meetings, the details of the intervention and the supporting documents are finalized.
5	13 and 20 December 2023 *	(1) Validate the intervention manual’s format; (2) Discuss future implementation; (3) Address necessary support; and (3) Identify anticipated obstacles.	Final discussions on materials and activities, implementation planning, support needs, and identification of potential obstacles.	N/A
6–8	19 March 2024 (CHUSJ) 14 May 2024 (CHULQ) 14 June 2024 (CHUS)	Pre-implementation meetings with each center to prepare program implementation. Each session focused on tailoring the intervention to center-specific contexts, defining staff roles, and planning for collaboration and communication.	Center-specific discussions covered implementation strategies, role assignments, logistical planning, coordination, and communication protocols. Each center’s unique needs and challenges were addressed to ensure smooth integration of the program.	Confirm final preparations for implementation within each center and provide any last-minute feedback to the program’s manual and worksheets.

* Two dates were offered due to a strike.

## Data Availability

The data presented in this study are available on request from the corresponding author. The data are not publicly available due to privacy and ethical restrictions.

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
