# Peer review of "A Multi-Site Refinement Study of Taking Back Control Together, an Intervention to Support Parents Confronted with Childhood Cancer"

_curroncol, 2025, doi:10.3390/curroncol32050253_

Round 1
Reviewer 1 Report
Comments and Suggestions for Authors
Dear authors,
You have accurately described the pathway you followed to ameliorate your intervention; this can be of inspiration to researchers that want to develop a similar project.
Title: I am not confident with the wording "refinement study". Is it a common definition or can it be substitute with a more used one?
Line 154: Could you please specify who conducted the discussion?
Table 1, meeting 1: Could you please add a brief explanation of what are the “Canoe trip” and “Parents’ Testimony”?
Table 1, meeting 2: Could you please add a brief explanation of what is “Marie’s story”?
Line 184: Here you refer to "predefined themes". Could you please list them?
Line 340: You named this item as “Marie’s story” in table 1 while here you named it as ““Marie’s case”. Is it intentional or is it an oversight?
Line 385: Could you please specify which kind of medical conditions make it inappropriate?
Table 4, Facilitator to implementation 4.16 and 4.17: What do you mean for “VIE 1.0” and “VIE”? You should explain the acronym.
Line 463: You should delete (TBCT) since you have already explained this acronym in the previous part of the text.
Author Response
Thank you for reviewing our manuscript. We are pleased to hear your largely positive feedback supporting our study's contribution to the empirical literature. We have considered each of your suggestions and comments, and we have modified our manuscript accordingly. Please find the detailed responses below and the corresponding revisions/corrections highlighted in yellow in the re-submitted files.
Comment 1: Title: I am not confident with the wording “refinement study.” Is it a common definition or can it be substitute with a more used one?
Response 1: Thank you for your comment. We used the terminology from the ORBIT model, which guided our study design. As this is a Phase 1b study, the term “refinement study” is consistent with the ORBIT model and refers to the iterative process of optimizing an intervention before pilot testing.
Comment 2: Line 154: Could you please specify who conducted the discussion?
Response 2: Thank you for your suggestion. We added a clarification to specify that two psychologists (SS and DO) conducted the discussion groups. This modification was made on line 152 of page 4 of the manuscript.
Comment 3: Table 1, meeting 1: Could you please add a brief explanation of what are the “Canoe trip” and “Parents’ Testimony”?
Response 3: As suggested, we added brief explanations in Table 1, Meeting 1 of page 5 of the manuscript to indicate the following: Canoe Trip, a short, animated video used as a metaphor to illustrate problem-solving steps, and Parents’ Testimony, a video featuring real parents sharing their experiences with their child’s diagnosis.
Comment 4: Table 1, meeting 2: Could you please add a brief explanation of what is “Marie’s story”?
Response 4: As suggested, we added a brief explanation in Table 1, Meeting 2 of page 5 of the manuscript to indicate the following: Marie’s Story, a written story that thoroughly demonstrates the six problem-solving steps taught in the intervention, serving as a concrete example to help parents understand and apply the method.
Comment 5: Line 184: Here you refer to “predefined themes.” Could you please list them?
Response 5: Thank you for your comment. The predefined themes are presented later in the manuscript (lines 200-245 of pages 7 and 8 of the manuscript). We added the following on lines 188-189 of page 7 of the manuscript to direct readers to where the predefined themes are detailed: themes are detailed in the sections outlining the analyses for each objective.
Comment 6: Line 340: You named this item as “Marie’s story” in table 1 while here you named it as ““Marie’s case”. Is it intentional or is it an oversight?
Response 6: Thank you for pointing this out. It was an oversight. We modified the manuscript to consistently use the same terminology – that is, “Marie’s story”. These changes were made on lines 344 and 371, as well as in Table 3, citation 3.14 of page 13 of the manuscript.
Comment 7: Line 385: Could you please specify which kind of medical conditions make it inappropriate?
Response 7: Thank you for your comment. We have added clarification to specify that certain medical conditions, such as advanced cancer, i.e., a condition that cannot be cured with standard treatment, may make it inappropriate for families to fully engage in the intervention. This modification was made on lines 390-391 of page 14 of the manuscript.
Comment 8: Table 4, Facilitator to implementation 4.16 and 4.17: What do you mean for “VIE 1.0” and “VIE”? You should explain the acronym.
Response 8: As suggested, we clarified the acronym by adding the following explanation: Valorization, Implication, and Education (VIE): the overarching initiative under which TBCT was developed. VIE 1.0 and VIE are the same; they refer to the first version of the intervention. This modification was made in Table 4, citation 4.16 of page 17 of the manuscript. We included a reference to the VIE project for readers to consult on lines 434-435 of page 15 of the manuscript. Finally, we also ensured that VIE was added to the list of abbreviations in the manuscript. This modification was made on line 617 of page 21 of the manuscript.
Comment 9: Line 463: You should delete (TBCT) since you have already explained this acronym in the previous part of the text.
Response 9: As suggested, we have removed TBCT in brackets in the discussion section on line 466 of page 18 of the manuscript.
Reviewer 2 Report
Comments and Suggestions for Authors
It was a complete joy to read this manuscript. It was comprehensive and insightful and I can imagine it will be a useful resource for other teams in their efforts to implement interventions.
As part of the background you could have mentioned the importance of this work for not only the parents and child affected by cancer but also siblings.
The issue of cultural representation was addressed as a potential limitation and avenue for research. Other aspects in relation to diversity which could have been addressed include socio-economic barriers (although travel was noted and people from certain geographical regions) and levels of literacy.
It is challenging to find areas to improve on given that this manuscript is of a high quality and has been carefully considered and written.
Author Response
Thank you for reviewing our manuscript. We are pleased to hear your largely positive feedback supporting our study's contribution to the empirical literature. We have considered each of your suggestions and comments, and we have modified our manuscript accordingly. Please find the detailed responses below and the corresponding revisions/corrections highlighted in yellow in the re-submitted files.
Comment 1: As part of the background you could have mentioned the importance of this work for not only the parents and child affected by cancer but also siblings.
Response 1: Thank you for your suggestion. We included the following information to highlight the effects of parental distress on siblings of children with cancer: As a result, siblings of children with cancer may undertake more household responsibilities, restricting their involvement in extracurricular and social activities [6]. […] Heightened parental distress may, in turn, make parents physically and emotionally unavailable, impacting the entire family unit [6]. This addition underscores the importance of this work for siblings of children with cancer. This modification was made on lines 67-73 of page 2 of the manuscript.
Comment 2: The issue of cultural representation was addressed as a potential limitation and avenue for research. Other aspects in relation to diversity which could have been addressed include socio-economic barriers (although travel was noted and people from certain geographical regions) and levels of literacy.
Response 2: As suggested, we included the following sentence on lines 565-567 of page 20 of the manuscript to address other aspects in relation to diversity: We also recognize the need to represent sexually diverse couples and consider how socio-economic barriers and levels of literacy may influence parents’ experiences navigating their child’s cancer journey.